# Electronic Games, Television, and Psychological Wellbeing of Adolescents: Mediating Role of Sleep and Physical Activity

**DOI:** 10.3390/ijerph18168877

**Published:** 2021-08-23

**Authors:** Asaduzzaman Khan, Nicola W. Burton

**Affiliations:** 1School of Health and Rehabilitation Sciences, The University of Queensland, Brisbane, QLD 4072, Australia; 2School of Applied Psychology, Griffith University, Mt Gravatt, QLD 4122, Australia; n.burton@griffith.edu.au; 3Menzies Health Institute Queensland, Griffith University, Gold Coast, QLD 4222, Australia

**Keywords:** mental health, screen use, exercise, young people, mediation analysis

## Abstract

This study investigated the associations between two common recreational screen activities and the psychological wellbeing of adolescents, and whether this association was mediated by sleep duration or physical activity frequency. This study used nationally representative cross-sectional survey data from 2946 adolescents (mean age 16.9 [0.38] years; 49% female) in the Longitudinal Study of Australian Children (LSAC). Adolescents provided information on daily time spent for each of the following: playing electronic games and watching television, time of sleep onset and wakeup, and number of days/week doing ≥60 min/day of physical activity. Psychological wellbeing was assessed by the Strengths and Difficulties Questionnaire (SDQ). Generalized estimating equations were used to examine the associations, and a contemporary multiple mediation analysis was used to examine the mediation effects. One fifth (20%) of adolescents were categorized as having poor wellbeing (SDQ total ≥17) with a significant sex difference (males: 16%; females: 24%; *p* < 0.001). Playing electronic games was inversely associated with psychological wellbeing for both male and female adolescents (*p* < 0.001). Watching television was inversely associated with psychological wellbeing for female adolescents (*p* < 0.001). Sleep duration and physical activity frequency were found to partially mediate the relationships between playing electronic games and the psychological wellbeing of male and female adolescents. Physical activity frequency partially mediated the association between television watching and wellbeing among female adolescents. Longitudinal studies are required to determine the causal pathway between screen-based activities and the wellbeing of adolescents, and to inform intervention strategies.

## 1. Introduction

Globally, as many as 20% of children and adolescents have one or more mental health conditions [1]. A national survey among US adolescents demonstrated a significant increase in self-reported depressive symptoms by about 20% from 2012 to 2018 [2]. Another study among children and young people in England demonstrated that 12.8% had at least one mental health condition in 2017 [3]. These conditions can have adverse implications for developmental processes, social interactions, educational attainment, and living productively [1] and can continue and become more severe later in life. This high prevalence of poor mental health and the implications for future wellbeing highlights the need for management strategies. However, approximately one third of adolescents with emotional or behavioral difficulties will not have used associated services [4]. Two thirds of adolescents will, however, use lifestyle behaviors to help manage problems [4]. There is, therefore, a need to identify how specific lifestyle behaviors are linked with psychological wellbeing.

There is increasing evidence that prolonged screen-based activities are common among adolescents, and associated with adverse mental health consequences, including depressive symptoms and poor quality of life [5]. Earlier research has reported positive associations between computer use, video games, and watching television (TV) with psychological difficulties including social anxiety, depression, and loneliness [6]. This association may vary by type of screen activity. One study demonstrated that high television watching, but not computer use, was positively related to depression in adolescents [7]. In contrast, another study reported that playing video games and computer use, but not television watching, were positively related to severe depressive symptoms, and only playing video games was positively related to severity of anxiety symptoms [8]. Other research with Australian adolescents indicated small adverse effects of screen time on psychosocial health, with considerable variation based on the type of screen time (e.g., passive, social, educational, interactive) [9].

When assessing the potential impact of screen time by gender, the type of mental health indicator may also be important to consider. Females are more vulnerable to depression than males, and males are more vulnerable to impulsivity than females [10]. Gaming has previously been associated with negative social outcomes in females but not males [11], and aggression in males but not females [12]. Therefore, depending on the outcome used, some previous studies of screen time and mental health may have created a sex bias and limited the understanding of potential associations with screen activities.

The inverse association between screen time and mental health may be mediated by other lifestyle behaviors. Adolescents with high screen time can self-impose “voluntary sleep restriction”, which can have a significant adverse impact on sleep patterns and quality [13]. Hypothesized mechanisms to explain the inverse relationship between screen time and sleep are that physiological and psychological arousal from the online material may delay or interrupt sleep, or prolonged exposure to blue light from the screen may suppress the level of sleep hormone melatonin [13]. Insufficient quality and quantity of sleep have been repeatedly related to the poor psychological health of adolescents [14]. A multi-country study among European adolescents demonstrated that the relationship between screen time and psychological symptoms was mediated by sleep duration and sleep onset difficulties [15]. However, that study assessed overall screen time combining academic (e.g., computer use for homework) and recreational (e.g., television, computer games) activities, which may have attenuated relationships if the psychological impact of screen time differed by type of use.

Physical activity could also mediate the inverse association between screen time and mental health. Physical activity has previously been positively associated with adult mental health [16,17], and the evidence for its benefits for adolescent mental health is promising [18,19]. Physical activity can contribute to mental health via a range of biopsychosocial mechanisms including increased levels of serotonin and endorphins which are involved in regulating and improving mood, reduced skeletal muscle tension, as a distraction from and outlet for frustrations, improved skills which boost self-esteem, and providing mastery experiences and social opportunities. The evidence on the relationship between screen time and physical activity is mixed with some studies reporting an inverse relationship [20,21], and others showing no association [22,23]. During adolescence, screen behaviors tend to increase, and physical activity tends to decrease, specifically in females [24]. Some research has identified independent relationships of screen time and physical activity with the mental wellbeing of adolescents [24], while other studies suggest that low physical activity and prolonged screen time interact to worsen psychological ill-being among adolescents [6]. Given this emerging and mixed evidence, there is a need to better understand the interrelationships among these three variables.

In recent years, Australia has experienced a sharp increase in poor mental health among adolescents. Mental ill-being among adolescents has increased by approximately 22% over five years, from 18.7% in 2012 to 22.8% in 2016, with a higher burden in females than males (28.6% and 14.1%, respectively, in 2016) [25]. Given the high use of screens for recreational pursuits, such as watching television and gaming, by Australian adolescents [9], and the interrelationships between different screen use types and psychosocial health [20,23,26], it is important to understand how screen-based activities are linked with psychological wellbeing. The aim of this study was, therefore, to examine the association between both playing electronic games and watching television, and the psychological wellbeing of Australian adolescents, and whether this was mediated by sleep or physical activity. Given previously demonstrated sex differences in screen use for gaming and television [27], physical activity [28], sleep [29], and wellbeing [28,30], the analysis was performed separately for male and female adolescents.

## 2. Materials and Methods

### 2.1. Study Population

Data for this study were from wave 7 of the kindergarten (K) cohort of the Longitudinal Study of Australian Children (LSAC), which was conducted in 2016 (details elsewhere [31]). LSAC commenced in 2004 using a cross-sequential design and Medicare Australia (the national universal health insurance scheme) enrolment database as its sampling frame and collects data every two years from a nationally representative sample of children and their parents. A two-stage cluster sampling design was used with stratification by state and then by major metropolitan centers versus others. First, postcodes were randomly selected and then there was a random selection of in-scope children within each selected postcode. The postcode of each study participant is linked to the “Statistical Area 2” (SA2) area identifier, which is developed by the Australian Bureau of Statistics. Survey weights for each wave were calculated considering the selection probability of each child, and were adjusted for non-response, loss to follow-up, and benchmarked to population numbers in major categories of the population of children born in 2004. LSAC wave 7 data were used for the current study as these included the item to assess the number of days/week participants did ≥60 min/d of physical activity. Of the 3089 adolescents aged 16–17 years, 143 were dropped from the analysis due to non-response on the outcome measure of wellbeing, resulting in the final analytical sample of 2946 participants.

### 2.2. Outcome Measure

Adolescents’ psychological wellbeing was measured using the Youth Self-Report version of the Strengths and Difficulties Questionnaire (SDQ) [32]. The SDQ comprises 25 items with three response options: 0 for ‘not true’, 1 for ‘somewhat true’, and 2 for ‘certainly true’. The SDQ total is derived by summing the scores on 20 items from subscales assessing emotional symptoms, conduct problems, hyperactivity-inattention, and peer relationship problems [33]. This SDQ total ranges from 0 to 40 with higher scores suggesting poorer psychological wellbeing, and a score of 17 or more indicating a high level of psychological symptoms. The SDQ scores have good evidence of validity among Australian children [34].

### 2.3. Exposures

Four items were used and asked respondents how many hours and minutes on a typical: (a) weekend day do you play electronic games?; (b) weekday do you play electronic games?; (c) weekend day do you watch TV programs or movies at home?; and (d) weekday do you watch TV programs or movies at home? Weekdays and weekend days were weighted at 5:2 to generate average daily time spent on electronic games and TV separately.

### 2.4. Potential Mediators

Participants were asked four items about sleep onset and wake-up time: (a) “About what time do you fall asleep on a usual school night?”, (b) “About what time do you wake up in the morning on a usual school day?”, (c) “About what time do you fall asleep on the nights when you do not have school the next day?”, and (d) “About what time do you wake up in the morning on the days when you do not have school?”. Responses to each question were used to derive an average measure of sleep duration on a typical night by combining school day and non-school day sleep, weighted at 5:2. 

Physical activity was assessed with one item: “About how many days each week do you do at least 60 min of moderate or vigorous physical activity ? (This is all the time you spent in activities that increased your heart rate and made you breathe hard)”. Response options ranged from 0–7 days/week. This single item is widely used to assess physical activity in children and adolescents, and has acceptable psychometrics [35].

### 2.5. Potential Confounders

A list of covariates was considered including sex (male/female), age, area of residence (metropolitan/non-metropolitan), language spoken at home (non-English/English), cigarette smoking, and alcohol intake. A portable laser stadiometer (Invicta Plastics, Leicester, UK) and Tanita body fat/weight scales were used to measure height (cm) and weight (kg) of the study participants, and body mass index (BMI) z-scores were computed. Family socio-economic position (SEP) was assessed using annual family income, and both parents’ employment status and education. Socio-economic Indexes for Areas (SEIFA) were used to measure neighborhood socio-economic status. The inclusion of these variables was supported by earlier research on physical activity, screen time, and mental wellbeing [33,36].

### 2.6. Statistical Analyses

Descriptive statistics (e.g., means, standard deviations (SD), proportions) were used to summarize the key variables of interest. Of the list of covariates, SEIFA and smoking were considerably associated with SEP and alcohol intake, respectively, and as such, SEIFA and smoking were excluded from the analysis to avoid multicollinearity.

We used generalized estimating equations (GEE) to assess the total and direct associations between electronic gaming and watching television and psychological wellbeing, after adjusting for the revised set of covariates. GEE was used to considering the nested structure of the data (i.e., adolescents nested within statistical area 2 [SA2]).

We then followed the Hayes’ two-step procedure [37] to test the mediating role of moderate or vigorous physical activity (MVPA) frequency and sleep duration on the relationship between electronic gaming and watching television and psychological wellbeing. This multiple mediation procedure requires: (a) significant relationships between each of electronic gaming and watching television (exposures), and sleep duration and MVPA frequency (mediators); and (b) significant relationships between sleep duration and MVPA frequency (mediators) and psychological wellbeing (outcome) after adjusting for electronic gaming and watching television (exposures). A mediation effect is present if the indirect effect of electronic gaming/watching television is significant. We used the PROCESS macro for SPSS to perform the multiple mediation analysis with bias-corrected bootstrap 95% confidence intervals (CIs) in 10,000 bootstrapping resamples (Model 4, Process Macro). When zero is not within the 95% CIs, the indirect effect is considered significant, suggesting the presence of a mediating effect in the association. We repeated the analysis to examine the robustness of the findings (sensitivity analysis) where the SDQ total was dichotomized to indicate the presence (yes: 17–40) or absence (no: 0–16) of poor psychological wellbeing. The significant difference in average SDQ score between male [10.5 (5.8)] and female [11.8 (5.9)] adolescents (*p* < 0.001) confirmed the a priori decision to run the analyses separately for male adolescents and female adolescents. The significance level was set at 5%.

## 3. Results

Study participants were aged 16–17 years, 48.9% were female, and 61% were living in metropolitan areas (Table 1). Overall, 20% of adolescents were categorized as having poor psychological wellbeing (SDQ total ≥17), which included 16.4% of males and 23.8% of females (*p* < 0.001).

### 3.1. Associations between Screen Activities, Physical Activity, and Sleep Duration [a]

After adjusting for confounders, time playing electronic games was significantly associated with both MVPA frequency [male adolescents: −0.19 (−0.27, −0.12); females: −0.20 (−0.31, −0.09)] and sleep duration [males: −0.07 (−0.10, −0.04); females: −0.07 (−0.12, −0.03)] (Figure 1). As shown in Figure 2, time watching television was significantly associated with MVPA frequency [−0.16 (−0.23, −0.09)] but not sleep duration [0.02 (−0.02, 0.06)] among female adolescents. Among male adolescents, time watching television was not associated with MVPA frequency or sleep duration (Figure 2).

### 3.2. Associations between Physical Activity, Sleep Duration, and Psychological Wellbeing [b]

After adjusting for confounders including playing electronic games, MVPA frequency was significantly associated with psychological wellbeing for both male [−0.21 (−0.37, −0.05)] and female [−0.39 (−0.60, −0.17)] adolescents (Figure 1). Sleep duration was also significantly related to psychological wellbeing for both male [−0.75 (−1.11, −0.38)] and female [−0.99 (−1.41, −0.58)] adolescents.

When controlled for confounders including watching television, MVPA frequency was significantly associated with the psychological wellbeing for male [−0.27 (−0.42, −0.12)] and female [−0.39 (−0.57, −0.21)] adolescents. Sleep duration was also significantly associated with the psychological wellbeing of male [−0.78 (−1.12, −0.43)] and female [−0.92 (−1.27, −0.56)] adolescents (Figure 2).

### 3.3. Associations between Electronic Gaming/Watching Television and Psychological Wellbeing [c]

The direct effects of electronic gaming and watching television on psychological wellbeing were assessed using GEEs. The associations, without considering mediator variables, and after adjustment for confounders, are shown, by sex, in Table 2. All association estimates (i.e., regression coefficients) were significant except the estimate between watching television and the psychological wellbeing of male adolescents.

The total effects of electronic gaming and watching television on psychological wellbeing were assessed using GEEs. The associations between electronic gaming/watching television and psychological wellbeing, adjusted for MVPA frequency, sleep duration, and the other confounders, by sex are shown in Table 2. All direct effect estimates (i.e., regression coefficients) were significant except for the estimate between watching television and the psychological wellbeing of male adolescents.

The indirect effects of electronic gaming and watching television on psychological wellbeing were assessed using a mediation analysis. The indirect effects of MVPA frequency and sleep duration on the relationships between electronic gaming/watching television and psychological wellbeing, by sex, are shown in Figure 1 and Figure 2. Both MVPA frequency and sleep duration did partially mediate the relationship between playing electronic games and the psychological wellbeing of both male and female adolescents (Figure 1). Together, MVPA frequency and sleep duration mediated about 22% of the total effect in male adolescents and about 15% of the total effect in female adolescents. As shown in Figure 2, only MVPA frequency partially mediated (about 12% of the total effect) the relationship between watching television and the psychological wellbeing of female adolescents. Neither MVPA frequency nor sleep duration mediated the relationship between watching television and the psychological wellbeing of male adolescents (Figure 2).

Sensitivity analyses showed similar results to the total, direct, and indirect effects when the categorical SDQ was used instead of the continuous SDQ (data not shown).

## 4. Discussion

The aim of this study was to examine the association between two common recreational screen activities and the psychological wellbeing of adolescents, and if these associations were mediated by sleep or physical activity. In the current study, 20% of adolescents were categorized as having poor psychological wellbeing, which included 16% of the males and 24% of the females. Time spent on electronic games was positively associated with poor wellbeing in males and females, and this was partially mediated by both sleep duration and physical activity frequency. Time watching television was positively associated with poor wellbeing among females only, and this was partially mediated by physical activity frequency only. This evidence can contribute to an understanding of the interrelationships among modifiable lifestyle factors and adolescent wellbeing, and potentially inform intervention strategies and public health advice.

Results from the current study demonstrate that both types of recreational screen activities were inversely associated with psychological wellbeing among adolescents, which is consistent with previous studies on the adverse psychological impacts of screen time [6], video games [8], and watching television [7]. Our study builds on the evidence in this area by showing that this association differed between screen type and sex. Time spent on electronic games was inversely related to wellbeing in males and females; however, watching television was inversely associated with wellbeing only among females. The current study used a broad assessment of psychological wellbeing that included emotional (e.g., worry, unhappiness, nervousness), attentional (e.g., restlessness, over-activity, distractibility, thinking before acting), conduct (e.g., temper loss, fighting, lying, cheating, stealing), and relationship (e.g., likeability, being bullied, solitariness, having a good friend) factors. This multidimensional assessment increased the potential sensitivity of our study given previous research demonstrating gender differences in vulnerability to specific types of poor mental health [28,30].

The relationship between time spent watching television and poor wellbeing in only the female adolescents may reflect the general nature and specific content of this screen activity, as well as sex differences in vulnerability. Previous research has made the distinction between passive (e.g., television watching) and mentally active (e.g., electronic gaming) sedentary behaviors, noting that passive sedentary behaviors increase the risk of depression [38], which females are more vulnerable to than males [30]. Television content can provide exposure to idealized images that prompt social comparisons and body image dissatisfaction, which female adolescents are more vulnerable to than males [39], and which may precipitate depression symptoms. Looks are important to women who frequently watch reality television [40].

Our results indicate that the inverse relationships between both types of recreational screen time and psychological wellbeing were partly mediated by sleep duration and physical activity frequency. Electronic gaming was inversely related to both sleep duration and physical activity frequency. Watching television was inversely related to only physical activity frequency, and only among female adolescents. Both electronic gaming and television may be seen by adolescents as more attractive recreational options than physical activity. A review study provides some support that physical activity and electronic media use compete in adolescents [41]. Among female adolescents, negative physical activity attitudes related to required exertion, (low) ability, and feminine stereotypes [42] may make screen activities, such as gaming and television, more attractive, and the social, relational, and appearance-focused content of television may be particularly appealing. In contrast with other research demonstrating that sleep problems can mediate the deleterious relationship of passive sedentary behavior with depression [43], our study found sleep duration did not mediate the association between watching television and wellbeing.

It is important to note that sleep duration and physical activity frequency only partially mediated the relationships between both types of recreational screen use and psychological wellbeing in the current study. Together they accounted for approximately 21% of the total effect in males and approximately 15% of the total effect in females for electronic gaming. Alone, physical activity frequency inversely accounted for approximately 12% of the total effect for watching television, and this was only for females. Neither sleep duration nor physical activity frequency inversely mediated the relationship between watching television and wellbeing among males.

Other underlying mechanisms, therefore, contribute to the relationship between these two types of screen activities and poor psychological wellbeing. Both electronic gaming and television can be solitary activities, thereby reducing the social opportunities required for brain and behavioral development associated with self-identity, self-referential processing, executive control, and mentalizing, which in turn are implicated in depression and empathy [44]. These screen activities may therefore prevent adolescents from developing and practicing functional coping strategies, making friends, and accessing social support. Gaming has been associated with neurobiological changes such as impaired motor response inhibition, reduced white matter tracts which is linked to executive functions (e.g., self-control), reduced dopaminergic transporters which can contribute to depression, and increased functional connectivity which can influence affective and anxiety symptoms [45]. Gaming is also associated with impulsive decision making and a preference for immediate gratification [45], which can adversely impact behavioral conduct and peer relations. Television can create or reinforce unrealistic expectations and maladaptive attitudes [46], which may compromise emotional wellbeing and social relations. More work is needed, therefore, to understand the physiological and psychological factors other than sleep duration and physical activity frequency that underlie the positive relationship between prolonged recreational screen activities and poor wellbeing.

This study has several strengths including a large nationally representative sample, a standardized multidimensional measure of psychological wellbeing that assessed emotional, cognitive, behavioral, and social symptoms and provided more sensitivity than a unidimensional measure, GEE modelling to take into account the clustering of data, and adjustment for multiple potential confounders. The analyses were stratified by sex, which offered a better insight into the sex-stratified mechanisms of the relationships. However, the present study was based on adolescents across a narrow age range (16–17 years), so results may not be generalizable to other age groups. All data were assessed using self-report, which is vulnerable to comprehension issues and subjective interpretation, e.g., watching television could refer to using a phone or tablet, as well as a television unit, and may have been answered differently across respondents. Self-reported data are also susceptible to recall and social desirability bias. The mediators (watching television and gaming) and covariates were measured using self-reported items without established psychometrics in the Australian adolescent population. The assessment of screen use focused on electronic games and watching television and did not include other common recreational screen-based activities such as social media use, which may have different associations with wellbeing and the proposed mediating behaviors. Sleep duration was measured using time of sleep onset and wake-up time and may not represent uninterrupted sleep or sleep quality which are also important for wellbeing. Despite providing data comparable with international behavioral guidelines, physical activity was assessed as number of days/week doing ≥60 min/day, which is more basic than other types of subjective or objective measures. A more sensitive measure, such as average daily duration, may have provided different results. As physical activity frequency data were only available in wave 7 of the LSAC study, only cross-sectional associations have been explored in the current analyses, which preclude any causal inferences about the relationships. Further research using longitudinal data is therefore recommended. The analyses were adjusted for a set of covariates available in the survey and may not have included other potential confounders.

## 5. Conclusions

Understanding the factors associated with poor psychological wellbeing in adolescents is important to inform intervention strategies targeting multiple behaviors and directions for future research. In this study, there was a positive relationship between time spent on electronic gaming and poor psychological wellbeing among female and male adolescents, which was mediated in part by reduced sleep duration and infrequent physical activity. Among female adolescents only, there was also a positive relationship between watching television and poor psychological wellbeing, which was mediated in part by infrequent physical activity. These results suggest that managing time across of these lifestyle activities may help to promote adolescent psychological wellbeing. More longitudinal studies are required to explore the directionality of relationships among these common lifestyle factors and to understand the underlying mechanisms, which in turn can inform effective interventions for the emotional, cognitive, and social wellbeing of adolescents.

## Figures and Tables

**Figure 1 ijerph-18-08877-f001:**
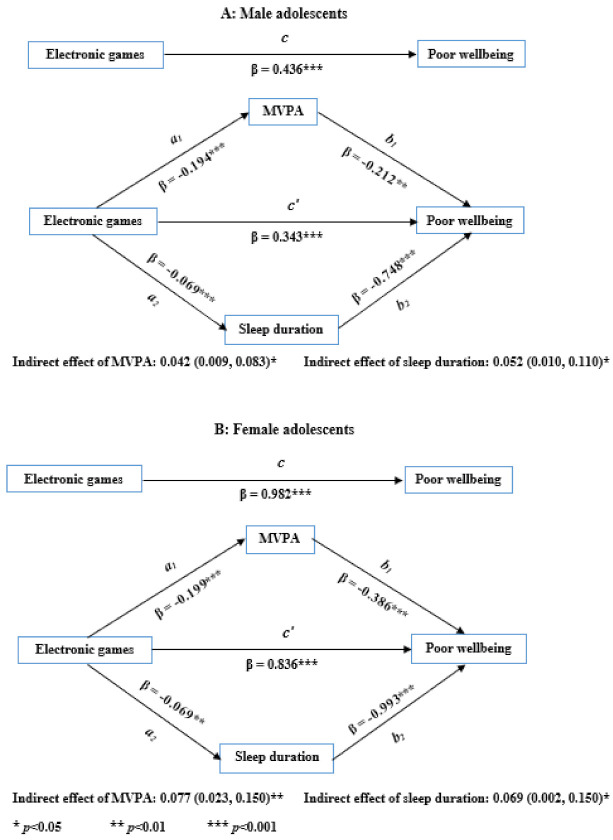
Models of moderate to vigorous physical activity (MVPA) frequency and sleep duration mediating the association between time playing electronic games and psychological wellbeing of (**A**) male and (**B**) female adolescents, 2016.

**Figure 2 ijerph-18-08877-f002:**
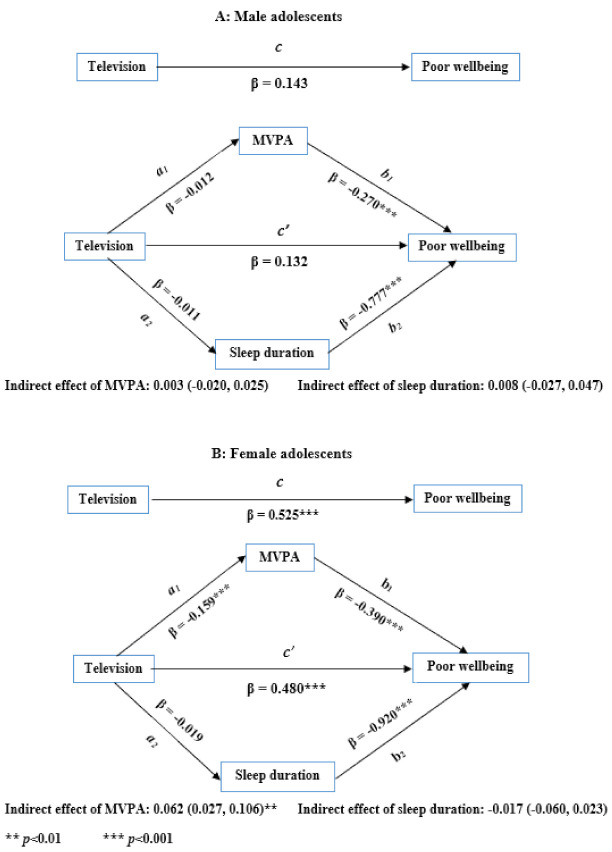
Models of moderate to vigorous physical activity (MVPA) frequency and sleep duration mediating the association between time watching television and psychological wellbeing of (**A**) male and (**B**) female adolescents, 2016.

**Table 1 ijerph-18-08877-t001:** Characteristics of the study participants, 2016 (*n* = 2946).

Characteristics	All Sample
Age, years [M (SD)]	16.9 (0.4)
Female [%]	48.9
Area of residence [%]	
Metropolitan	61.3
Non-Metropolitan	38.7
Language spoken at home [%]	
English	90.5
Not-English	9.5
Alcohol use [%]	
None	42.3
Some	57.7
Cigarette smoking [%]	
None	78.6
Some	21.4
Average time watching television (h/d) [M(SD)]	2.2 (1.7)
Average time playing electronic games (h/d) [M(SD)]	1.5 (1.8)
Average MVPA ^ɸ^ frequency days/wk [M(SD)]	2.36 (2.13)
Average sleep duration (h/d)	8.4 (1.2)
Average wellbeing score [M (SD)]	11.1 (5.9)
Males	10.5 (5.8)
Females	11.8 (5.9)
Poor psychological wellbeing^¥^ [%]	20.0
Males	16.4
Females	23.8

^ɸ^ Moderate-to-vigorous physical activity ≥60 min/d. **^¥^** Derived by dichotomizing the SDQ total where 17–40 represents presence, and 0–16 represents absence, of poor psychological wellbeing.

**Table 2 ijerph-18-08877-t002:** The total and direct effects of electronic gaming and watching television on the psychological wellbeing of Australian male and female adolescents, 2016.

	Total Effects	Direct Effects
	β (95% CI)	β (95% CI)
Model 1a: Time playing electronic games—Males	0.44 (0.25, 0.63) ***	0.34 (0.15, 0.54) ***
Model 1b: Time playing electronic games—Females	0.98 (0.66, 1.30) ***	0.84 (0.52, 1.16) ***
Model 2a: Time watching television—Males	0.14 (−0.05, 0.34)	0.13 (−0.06, 0.33)
Model 2b: Time watching television—Females	0.52 (0.30, 0.75) ***	0.48 (0.26, 0.70) ***

*** Estimate was significant at *p* ≤ 0.001. β = regression coefficient, and 95% CI = 95% confidence interval. Direct effect models were adjusted for age, alcohol intake, language spoken at home, BMI z-scores, socio-economic position (SEP), and region of residence. Total effect models were adjusted for sleep duration and moderate-to-vigorous physical activity (MVPA) frequency in addition to age, alcohol intake, language spoken at home, BMI z-scores, socio-economic position (SEP), and region of residence.

## Data Availability

The LSAC data can be accessed upon request: https://growingupinaustralia.gov.au/.

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
