# Peer review of "Electronic Games, Television, and Psychological Wellbeing of Adolescents: Mediating Role of Sleep and Physical Activity"

_ijerph, 2021, doi:10.3390/ijerph18168877_

Round 1

Reviewer 1 Report

General comment: This study aimed to examine the association between each of playing electronic games and watching TV, and psychological wellbeing in Australian adolescents, and whether this was mediated by sleep or physical activity. To reach this objective is relevant and could help to better understand the associations between lifestyle behaviors and psychological wellbeing. However, I am concerned about the methodology used to reach this objective.

Major comments:

1. As this study population is 16- and 17-years adolescents, it seems an important limitation to collect data on screen activities without having data on social media activities, on which they spend a great part of their screen time. Without it, I feel like a piece of the puzzle is missing. If you have collected such information, please include it in the current manuscript.

2. Lines 131-135: Do this way of assessing MVPA allow to really assess the potential mediator role of physical activity? What about an adolescent who is performing 50 minutes/per day each day? Does he should be considered as "inactive" with a "0 day per week" score in this index? I know you acknowledge the limitation cause by this question's formulation, but, as we know that physical activity acts as a mental health protective factor even when performing below 60 minutes per day, I am wondering if this indicator may lead to "inaccurate" results.

3. The Results section is difficult to read. Different comments through this section should be carefully addressed (see main document in pdf). For example:

-A new variable is presented in Table 1 (Average MVPA). If you have that information on participants, it would be a better indicator of MVPA than the one used.

-The figures are offset and far from the text they refer to. It is unpleasant to read as we are always scrolling a lot to follow. Please consider repositioning the figures and even separate them to facilitate the reading of this section. Also, some data are missing in some figures.

-Lines 236-247: The presentation of indirect effects must be clarified (especially the link between text and figures).

4. Discussion (lines 286-325): Please discuss about the mechanisms explaining the mediating role of MVPA and sleep duration on the screen time and wellbeing association.

Minor comments:

Please see minor comments directly in the pdf.

Author Response

Reviewer # 1

General comment: This study aimed to examine the association between each of playing electronic games and watching TV, and psychological wellbeing in Australian adolescents, and whether this was mediated by sleep or physical activity. To reach this objective is relevant and could help to better understand the associations between lifestyle behaviors and psychological wellbeing. However, I am concerned about the methodology used to reach this objective.

Major comments:

  1. As this study population is 16- and 17-years adolescents, it seems an important limitation to collect data on screen activities without having data on social media activities, on which they spend a great part of their screen time. Without it, I feel like a piece of the puzzle is missing. If you have collected such information, please include it in the current manuscript.

Response: We agree with the reviewer that data on social media activities would have provided a much better snapshot of adolescents’ screen use. However, the LSAC survey, on which the current analysis is based, did not collect data on social media and hence it was beyond the scope of the current study to include social media in the analyses. This is why we titled the study to specifically indicate that it only included electronic gaming and television use (vs screen time). To emphasise this, we have now also revised the text throughout the manuscript to replace the more generic label “recreational screen time” with “electronic gaming” and “watching television” in relation to our analyses (although this term is kept in the introduction and discussion to accurately describe previous research). The non-inclusion of social media as a common other type of screen time has been acknowledged as a limitation of the study. (page 11, lines 385-86)

  1. Lines 131-135: Do this way of assessing MVPA allow to really assess the potential mediator role of physical activity? What about an adolescent who is performing 50 minutes/per day each day? Does he should be considered as "inactive" with a "0 day per week" score in this index? I know you acknowledge the limitation cause by this question's formulation, but, as we know that physical activity acts as a mental health protective factor even when performing below 60 minutes per day, I am wondering if this indicator may lead to "inaccurate" results.

Response: We agree that this is not the most sensitive measure of physical activity; however, LSAC only used a validated single item to assess the number of days each week respondents did moderate-to-vigorous physical activity (MVPA) ≥60 mins per day. This provides data comparable with Australian movement guidelines for children and adolescents aged 5-17 years to do at least 60 mins daily of MVPA, and the World Health Organisation recommendations of an average of 60 mins/day of MVPA across the week. To emphasise how MVPA was measured in the current study, we have revised the text to label the physical activity variable as “MVPA frequency” throughout the manuscript, and do not categorise or refer to participants as “active” or “inactive”. The potential limitations of this approach to measurement have also been considered in the discussion. (page 11, lines 389-96)

  1. The Results section is difficult to read. Different comments through this section should be carefully addressed (see main document in pdf). For example:

-A new variable is presented in Table 1 (Average MVPA). If you have that information on participants, it would be a better indicator of MVPA than the one used.

Response: We have addressed the comments provided (i.e., “Average MVPA” was replaced by “MVPA frequency” and revised the results text and tables to ensure consistency of variable labels throughout the manuscript including Table 1.  

-The figures are offset and far from the text they refer to. It is unpleasant to read as we are always scrolling a lot to follow. Please consider repositioning the figures and even separate them to facilitate the reading of this section. Also, some data are missing in some figures.

Response: We apologise for the missing data which was due to the PDF formatting of our WORD document. We have fixed the Figures and provided all relevant data in the revised manuscript.

-Lines 236-247: The presentation of indirect effects must be clarified (especially the link between text and figures).

Response: With the revised presentation of formatted Figures and added missing relevant data, the revised text is linked with the Figures 1 and 2.

  1. Discussion (lines 286-325): Please discuss about the mechanisms explaining the mediating role of MVPA and sleep duration on the screen time and wellbeing association.

Response: The potential mechanisms for the mediating role of sleep and physical activity between the two screen activities and wellbeing are provided in the introduction, so we are reluctant to repeat that information in the discussion. We indicated that

  • prolonged screen time (i.e., watching television, gaming) can adversely impact on sleep via voluntary sleep restriction, physiological and psychological arousal from online material, prolonged exposure to blue light suppressing melatonin, and insufficient sleep is linked to poor psychological wellbeing, and
  • screen time can increase, and physical activity decrease during adolescence, and that watching television or gaming may replace or be preferred to physical activity.

We have now also added to the introduction that physical activity is positively associated with wellbeing via a range of biopsychosocial mechanisms, such as increased levels of serotonin and endorphins which are involved in regulating and improving mood, reduced skeletal muscle tension, distraction from and outlet for frustrations, mastery experiences, improved physical skills which boosts self-esteem, and social opportunities. (page 2, lines 86-90)

Minor comments:

Please see minor comments directly in the pdf.

Response: We have addressed all the comments in the revised manuscript and added our responses for each of the comments in the PDF file (copy attached for info).

Reviewer 2 Report

In the abstract change “ Prospective studies are required” to experimental.

Methods

Race/ethnicity should be used as a cofounder. If not please include in the limitations.

The validity and reliability of all the tools should be provided and cited.

Total citations should be kept to about 30, please consider.

Author Response

In the abstract change “ Prospective studies are required” to experimental.

Response: Thank you for the suggestion; we have replaced ‘prospective’ by ‘longitudinal’ as the latter seems to be more appropriate to us in this context.

Methods

Race/ethnicity should be used as a cofounder. If not please include in the limitations.

Response: We thank the reviewer for raising this. The analyses were adjusted for language spoken at home, which was considered a proxy indicator of race/ethnicity. 

The validity and reliability of all the tools should be provided and cited.

Response: Psychometric properties of the outcome measure (SDQ: wellbeing) and physical activity are reported or cited in the manuscript [Ref#, 34, 35, respectively]. We are not aware of psychometrics of the items used to assess time spent in gaming and watching television. Tools used to measure covariates in LSAC are widely used; however, their reliability within the Australia adolescent population is yet to be established. This has been acknowledged as a limitation in the revised manuscript.

Mediators (watching television and gaming) and covariates were measured using self-reported items without established psychometrics in the Australian adolescent population.” (page 11, lines 381-83)

Total citations should be kept to about 30, please consider.

Response: As this work includes literature related to five different areas of adolescent lifestyles (gaming, television, sleep duration, physical activity, wellbeing), we have been obliged to use a range of references. We consider that all the provided references are important and relevant to the text presented.  

Reviewer 3 Report

General Comments

In “Electronic games, television, and psychological wellbeing of adolescents: mediating role of sleep and physical activity”, the authors examined the associations between two common recreational screen activities and psychological wellbeing of adolescents, and whether this association was mediated by sleep duration or physical activity. Overall, I thought the article was well written and the analyses were appropriate for this research question. The introduction was clear and nicely set up the motivation for the analyses explored in this study. The analysis and results were clearly presented. I have only a few minor comments (please see specific comments below).

Specific Comments

Abstract

  • Line 15: In a formal questionnaire or as part of a larger survey?
  • If there is enough room, adding analysis (even woven into results) would be helpful.

Introduction

  • Line 54: Minor comment, but the motivation for exploring sex differences could almost be its own paragraph
  • Lines 61 and 75: These two paragraphs (where potential mediators are introduced) may benefit from some transitional phrasing as they sort of abruptly change focus.
  • Line 94: During the age range mentioned earlier, screen viewing, particularly for entertainment purposes, became more accessible with increasing availability of portable devices. Does "watching television" encompass this?

Methods

  • Line 103: Although details are available elsewhere, basic information regarding eligibility/recruitment population would be helpful here. Also, broadly, what was involved with the LSAC (specifically at Wave 7)?
  • Line 116: Does this include adolescents?

Results

  • Figure 1: The alignment appears a little off (at least in my proof) (e.g., in 1B, the title is hidden behind the figure).

Discussion

  • Line 302: This is a rather dense paragraph and I wonder if it can be broken up a bit. There is some redundancy with multiple sentences starting with "Gaming..."
  • Line 336: I am glad the authors bring up social media use, as this may be important influence on these mediating behaviors and the outcome and should be considered in future work.

Author Response

Reviewer # 3

In “Electronic games, television, and psychological wellbeing of adolescents: mediating role of sleep and physical activity”, the authors examined the associations between two common recreational screen activities and psychological wellbeing of adolescents, and whether this association was mediated by sleep duration or physical activity. Overall, I thought the article was well written and the analyses were appropriate for this research question. The introduction was clear and nicely set up the motivation for the analyses explored in this study. The analysis and results were clearly presented. I have only a few minor comments (please see specific comments below).

Specific Comments

Abstract

  • Line 15: In a formal questionnaire or as part of a larger survey?

Response: Adolescents provided the information as a part of a large Longitudinal Study of Australian Children (LSAC) survey. We have revised the text to provide a description of LSAC. (page 1, lines 15-17)

  • If there is enough room, adding analysis (even woven into results) would be helpful.

Response: Thank you for this suggestion. We have added the following sentence on analysis.

Generalised estimating equations were used to examine the associations, and a contemporary multiple mediation analysis was used to examine the mediation effects.” (page 1, lines 18-20)

Introduction

  • Line 54: Minor comment, but the motivation for exploring sex differences could almost be its own paragraph

Response: Thank you for this suggestion. We have revised the text to present sex differences as a separate paragraph (page 2, starting at line 59). 

  • Lines 61 and 75: These two paragraphs (where potential mediators are introduced) may benefit from some transitional phrasing as they sort of abruptly change focus.

Response: This text has been revised to include transitional phrasing and have a more consistent format between the two paragraphs, so as to lessen the abrupt change of focus.

  • Line 94: During the age range mentioned earlier, screen viewing, particularly for entertainment purposes, became more accessible with increasing availability of portable devices. Does "watching television" encompass this?

Response: We are unable to conclusively state if using portable devices (e.g., phone, tablet) for watching television was intended to be included in the assessment of “watching television programs or moves at home” but assume it could be. We have added this possible issue to the study limitations text.

“All data were assessed using self-report, which is vulnerable to comprehension issues and subjective interpretation e.g., watching television could refer to using a phone or tablet, as well as a television unit, and so may have been answered differently across respondents.” (page 11, lines 377-80) 

Methods

  • Line 103: Although details are available elsewhere, basic information regarding eligibility/recruitment population would be helpful here. Also, broadly, what was involved with the LSAC (specifically at Wave 7)?

Response: Thank you for this suggestion, we have added some details about LSAC, eligibility/recruitment of the sample, and why wave 7 data were used (page 3, lines 117-29).

  • Line 116: Does this include adolescents?

Response: No, the study included children aged 4-9 years.

Results

  • Figure 1: The alignment appears a little off (at least in my proof) (e.g., in 1B, the title is hidden behind the figure).

Response: We apologies for the PDF formatting which obscured some information. We have fixed the Figure positions and provided all relevant data in the revised manuscript.  

Discussion

  • Line 302: This is a rather dense paragraph and I wonder if it can be broken up a bit. There is some redundancy with multiple sentences starting with "Gaming..."

Response:  We have split this paragraph into two and revised the text to reduce the redundancy of multiple sentences starting with “gaming”.

  • Line 336: I am glad the authors bring up social media use, as this may be important influence on these mediating behaviors and the outcome and should be considered in future work.

Response: We appreciate the comment and have added the following in the revised manuscript.   

“Assessment of screen use focused on electronic games and watching television, and did not include other common recreational screen-based activities such as social media use, which may have different associations with wellbeing and the proposed mediating behaviours.” (page 11, lines 383-86)

Round 2

Reviewer 1 Report

Thank you for answering all the comments and modifying the manuscript accordingly. I have no further comment.